# Association of Equine Herpesvirus 5 with Mild Respiratory Disease in a Survey of EHV1, -2, -4 and -5 in 407 Australian Horses

**DOI:** 10.3390/ani11123418

**Published:** 2021-11-30

**Authors:** Charles El-Hage, Zelalem Mekuria, Kemperly Dynon, Carol Hartley, Kristin McBride, James Gilkerson

**Affiliations:** 1Centre for Equine Infectious Diseases, Faculty of Veterinary and Agricultural Sciences, The University of Melbourne, Parkville, VIC 3010, Australia; mekuria.3@osu.edu (Z.M.); kemperly@gmail.com (K.D.); carolah@unimelb.edu.au (C.H.); jrgilk@unimelb.edu.au (J.G.); 2Global One Health Initiative, Ohio State University, Columbus, OH 43210, USA; 3UNSW Medicine, University of New South Wales, Sydney, NSW 2052, Australia; kmcbride@kirby.unsw.edu.au

**Keywords:** gammaherpesvirus, horses, respiratory disease, equine herpesvirus 1, -2, -4, -5, equine influenza, quantitative PCR

## Abstract

**Simple Summary:**

Infectious respiratory diseases in horses represent a major health and welfare problem. Although equine influenza is well reported as a cause of respiratory disease in most continents, Australia is free of EIV despite an outbreak in two states in 2007. Horses in Victoria were tested to demonstrate proof of freedom from EIV, hence samples were able to be subsequently tested for this study with the knowledge that EIV was not present as a potential cause of any disease. The equine alphaherpesviruses, EHV1 and -4 are well known agents of equine respiratory disease. The gammaherpesviruses EHV2 and -5 on the other hand are often isolated from clinically healthy horses despite a known association in some disease processes. The consequences of infection with these enigmatic viruses remains unknown. The investigation of several hundred horses with and without respiratory disease provided valuable information in terms of association. The salient findings of this study determined that a large proportion of normal horses were positive for the gammaherpesviruses EHV2 and -5 using PCR methods. However, horses shedding EHV5 were more likely to have had signs of respiratory disease. Like EHV2, EHV5 is a gammaherpesvirus commonly found in horses: its significance is unclear, though it is closely related to the Epstein–Barr virus, the agent responsible for glandular fever in humans. These viruses are known to interfere with the immune response and have potentially wide-ranging effects on infected hosts. This study has added to our awareness of these equine herpesviruses and should stimulate further studies to determine exact causation and consequences of infection.

**Abstract:**

Equine herpesviruses (EHVs) are common respiratory pathogens in horses; whilst the alphaherpesviruses are better understood, the clinical importance of the gammaherpesviruses remains undetermined. This study aimed to determine the prevalence of, and any association between, equine respiratory herpesviruses EHV1, -2, -4 and -5 infection in horses with and without clinical signs of respiratory disease. Nasal swabs were collected from 407 horses in Victoria and included clinically normal horses that had been screened for regulatory purposes. Samples were collected from horses during Australia’s equine influenza outbreak in 2007; however, horses in Victoria required testing for proof of freedom from EIV. All horses tested in Victoria were negative for EIV, hence archived swabs were available to screen for other pathogens such as EHVs. Quantitative PCR techniques were used to detect EHVs. Of the 407 horses sampled, 249 (61%) were clinically normal, 120 (29%) presented with clinical signs consistent with mild respiratory disease and 38 (9%) horses had an unknown clinical history. Of the three horses detected shedding EHV1, and the five shedding EHV4, only one was noted to have clinical signs referable to respiratory disease. The proportion of EHV5-infected horses in the diseased group (85/120, 70.8%) was significantly greater than those not showing signs of disease (137/249, 55%). The odds of EHV5-positive horses demonstrating clinical signs of respiratory disease were twice that of EHV5-negative horses (OR 1.98, 95% CI 1.25 to 3.16). No quantitative difference between mean loads of EHV shedding between diseased and non-diseased horses was detected. The clinical significance of respiratory gammaherpesvirus infections in horses remains to be determined; however, this survey adds to the mounting body of evidence associating EHV5 with equine respiratory disease.

## 1. Introduction

Equine herpesviruses (EHVs) are common respiratory pathogens in equids. These viruses have serious health and welfare outcomes in horses and significant financial consequences worldwide [1,2,3,4]. Both the alphaherpesviruses EHV1 and -4 are transmitted by the respiratory route, although respiratory disease is more commonly attributed to EHV4. The clinical importance of the gammaherpesviruses EHV2 and -5 is less clear [5,6,7]. This lack of clarity may be attributed to the frequent detection of gammaherpesviruses in horses with and without clinical signs of disease, under both experimental and field conditions [8,9,10,11,12,13,14,15,16,17,18]. Although outbreaks of disease caused by alphaherpesviruses are commonly reported in horses, shedding from the respiratory tract is often of short duration, and usually only detected in a minority of the population [4,19,20,21,22,23,24]. Many studies have detected gammaherpesviruses in a large percentage of horses within a population, often with few clinical signs of disease [9,14,15,25,26]. Although the gammaherpesviruses are commonly detected in clinical samples from horses, there are differences in the frequency of detection of these two viruses. The relative prevalence of these viruses varies in different studies, with some studies showing higher detection of EHV2 than EHV5 [12,15,27], and others showing EHV5 as more prevalent [17,25,26,28,29,30]. Several studies since 2007 have reported an association between EHV5 detection and a pulmonary fibrotic condition of horses, equine multi-nodular pulmonary fibrosis (EMPF) [18,31,32,33,34].

Individual horses can be infected with multiple herpesvirus species [29,35,36,37,38,39,40]. It has been hypothesised that infection with equine gammaherpesvirus may result in immunosuppression and, consequently, increased susceptibility to new or reactivated infections. While equine gammaherpesviruses contain many potential immunomodulating genes, [41,42] and gammaherpesvirus-mediated immunosuppression has been demonstrated in other species [43,44], this has not been reported as extensively in horses.

The opportunity to sample diseased and clinically normal horses arose during Australia’s only recorded equine influenza (EI) outbreak in 2007. The outbreak was limited to states north of Victoria, which remained free of EI. Equine respiratory samples were collected in Victoria for EI exclusion. This formed a central part of the outbreak investigation to confirm that equine influenza virus (EIV) had not spread to Victoria and was required for horse movement permits.

The aim of the study was to examine the prevalence of four endemic equine respiratory herpesviruses, and to determine if there was any association between infection and clinical respiratory disease.

## 2. Materials and Methods

### 2.1. Study Population

The study population consisted of 407 horses in Victoria with and without clinical signs of respiratory disease during the Australian EI outbreak from August 2007 to January 2008. These horses were sampled for the purposes of EI exclusion if they (i) had clinical signs of respiratory disease, (ii) potentially had contact with infected horses or (iii) required movement clearances. Clinical signs of mild respiratory disease were recorded as one or more of the following signs: coughing, pyrexia (temperature >38.5 °C) and/or nasal discharge [45]. A total of 522 nasal swabs were collected from these horses for exclusion of EIV. Vaccination histories were not recorded. In Australia, EHV1 and -4 (Duvaxyn, EHV-1, 4, Zoetis P/L, Castle Hill, NSW, Australia) and *Streptococcus equi* sub-species *equi* (Equivac-S™, Zoetis P/L, Castle Hill, NSW, Australia) vaccines are commonly used [46] while EIV vaccination is not permitted in Australia, unless for export purposes.

### 2.2. Nasal Swabs

Nasal swabs were collected using swabs with a 15 cm wooden shaft and a cotton tip (Interpath Services, Heidelberg West, VIC, Australia 163KS01) [47]. Swab tips were placed into 5 mL of Brain Heart Infusion (BHI) broth-based viral transport medium (BHI 3.7% *w*/*v* in sterile distilled water with penicillin 5000 U/mL, gentamicin 0.1 mg/mL, streptomycin 5 mg/mL and 200 μg/mL fungizone (Sigma Healthcare, Rowville, VIC, Australia)). Following testing for EI, swabs in transport media were stored at −80 °C. 

### 2.3. Nucleic Acid Extraction from Nasal Swabs

Nucleic acid extraction from nasal swabs was performed using an automated system (X-tractor Gene, Qiagen) using the QIAamp^®^ Virus Biorobot 9604 Kit (QIAGEN, Germantown, MD, USA) according to the manufacturer’s recommendations (https://www.qiagen.com (accessed on 20 December 2020)). Virus culture supernatants were included as known positive control samples and were also extracted using this method [48]). Viruses used as positive controls were EHV1.438/77 [49], EHV4.405/76 [49], EHV2.86/67 [50] and EHV5.2-141 [51].

### 2.4. Quantitative PCR Assays

All quantitative PCR (qPCR) tests were performed in a Stratagene© MxPro Mx3000P real-time PCR machine (Agilent Technologies Inc., Santa Clara, CA, USA), and analysed with the machine’s software with cycle threshold values assigned using the default threshold algorithm. Standard curves were generated from cycle thresholds of samples with known virus concentrations and genome copy numbers.

EHV1 and EHV4 were detected in a multiplex Taqman assay with the primers and probes targeting the EHV1 glycoprotein H gene, and the EHV4 intergenic region between open reading frames 73 and 74 (Appendix A, Table A1). These were used in a 20 µL reaction containing Brilliant qPCR Multiplex master mix (Stratagene, Agilent Technologies Inc., Santa Clara, CA, USA), 200 nM of each forward and reverse primer and probes, 30 nM ROX reference dye and 5 µL of sample DNA. The reaction thermocycling conditions were 95 °C for 10 min, followed by 40 cycles of 95 °C for 30 s and 60 °C for 1 min. Samples with a Ct value of ≤35 were considered positive.

Equine herpesviruses 2 and 5 were detected in two separate qPCR assays using SYTO^®^ 9 (Invitrogen™, Thermo Fisher Scientific, Waltham, MA, USA) as a double stranded DNA-binding dye. Equine herpesvirus 2 primers were designed to the glycoprotein B gene and equine herpesvirus 5 primers were designed to the glycoprotein H gene of EHV5 (Table A1). Each 25µL reaction volume contained 2 µg/mL SYTO9, 0.2 U GoTaq (Promega Corporation. Madison, WI, USA), 300 nM of the appropriate forward and reverse primers (Table A1) and 1.5 mM MgCl_2_ in the GoTaq reaction buffer as recommended by the manufacturer (Promega Corporation, Madison, WI, USA). Thermocycling of reactions proceeded at 94 °C for 15 min, then 40 cycles of 94 °C for 15 s, 60 °C annealing for 30 s and 72 °C extension for 30 s. The melting curve analysis of each amplicon was analysed after one cycle of 95 °C for 1 min, 55 °C for 30 s and 95 °C for 30 s. Samples were considered positive if the melting temperature of the amplicon was within the range specified below and the cycle threshold was below 35 in the EHV2 [52], and 37 in the EHV5 [41] assays, respectively. The melting temperature of the amplicon was determined using four diverse EHV2 isolates [10,51], and three EHV5 isolates [51] and occurred within the range 79 to 81 °C for EHV2 and 80 to 82 °C for EHV5. Positive control viruses EHV2.86/67 and EHV5.2-141 and nuclease-free water were included for each 96-well extraction and PCR plate. 

### 2.5. Statistical Analysis

Comparisons of two proportions were determined by Fisher’s exact test. A two-sample t-test was used to compare mean quantitative cycles between samples from non-diseased and diseased horses. Any two-sided Student’s t-test with a *p* value less than 0.05 was considered to be significant. Logistic regression methods were utilised to test for interaction. Statistical analysis was performed using Stata 12.1 Windows software (StataCorp, College Station, TX, USA). 

## 3. Results

### 3.1. Stratification of Horses in Terms of Respiratory Disease

Of the 407 horses sampled, 249 (61%) were clinically normal, 120 (29%) presented with clinical signs consistent with mild respiratory disease and 38 (9%) horses had an unknown clinical history (Table 1). For instances where multiple samples were taken at a single time point, a horse was reported as infected if viral DNA was detected in any sample collected at that time. 

### 3.2. Equine Herpesvirus Infections 

#### 3.2.1. Equine Herpesvirus 1

Equine herpesvirus 1 was detected in three horses (Table 1). The viral loads detected in these samples ranged from 10^6.45^, 10^7.91^ and 10^9.22^ genome copies/mL of nasal swab. None of these horses exhibited any clinical signs of respiratory disease at the time of sampling. 

#### 3.2.2. Equine Herpesvirus 4

Five horses were EHV4 positive. Three of these horses were clinically normal when sampled. The highest EHV4 load was 10^8.45^ genome copies/mL nasal swab from a horse of unknown clinical status. There were insufficient data for a meaningful comparison of aphaherpesvirus shedding between diseased and normal horses. 

### 3.3. Equine Gammaherpesvirus Infections, EHV2 and -5

In total, 83 (20.4%) of the 407 horses sampled were EHV2 positive and 249 horses (61.2%) were positive by qPCR for EHV5 (Table 1). There were no differences between the mean viral load of EHV2 or EHV5 detected in diseased and non-diseased horses (Figure 1). There was, however, a statistically significant difference (*p* = 0.004) between the proportion of horses in the diseased group shedding EHV5 (85/120, 70.8%) compared to the proportion of horses in the non-diseased group that were shedding EHV5 at the time of sampling (137/249, 55%). The odds of respiratory disease in EHV5-positive horses were twice that of EHV5-negative horses (OR 1.98, 95% CI 1.25 to 3.16). The proportion of horses with detectable EHV2 was significantly higher in non-diseased horses (61/249, 24.5%) compared to the diseased group (18/120, 15.0%) (*p* = 0.042). The odds of EHV2-positive horses also exhibiting clinical signs of disease were approximately half that of EHV2-negative horses (OR 0.54, 95% CI 0.30 to 0.97).

### 3.4. Concurrent Equine Herpesvirus Infections 

Of the 407 horses sampled in this survey, 54 (13.3%) were shedding multiple equine herpesviruses (Table 1). Two of the three horses shedding detectable levels of EHV1 were concurrently shedding EHV4. Three of the five EHV4-positive horses were also shedding EHV5, and a fourth was shedding EHV2. One horse that was clinically normal was shedding EHV2, -4 and -5 concurrently. The horse shedding the highest EHV4 load of 10^8.45^ copies/mL nasal swab was also shedding 10^7.94^ copies/mL of EHV5; however, the disease status of this horse was unknown.

Fifty of the eighty three horses (60.2%) shedding EHV2 were also shedding EHV5; however, there was no greater likelihood of EHV5 detection in these horses compared to those without detectable EHV2 (199/324, 61.4%; *p* = 0.90). In addition, these co-infected horses were no more likely to exhibit signs of disease (14/46, 30.4%) than those shedding only EHV2 (4/33, 12.1%; *p* = 0.063) or EHV5 (71/176, 40.3%; *p* = 0.24). Logistic regression showed no correlation between dual EHV2 and -5 shedding and clinical signs of respiratory disease (*p* = 0.41). Hence, the association of EHV5 infection and increased likelihood of disease was not modified by the presence or absence of EHV2 infection.

## 4. Discussion

Equine herpesvirus infections were commonly detected in samples from the respiratory tract, irrespective of clinical disease status at the time of sampling. Approximately 40% of horses were shedding at least one herpesvirus at the time of sampling (Table 1). In total, 67.9% of horses with no obvious clinical disease were shedding detectable levels of at least one herpesvirus. Detection of the alphaherpesviruses in a small proportion of horses (2%, *n* = 8) contrasted markedly with the high frequency of shedding of the equine gammaherpesviruses (69.3%, *n* = 282). Although many clinically normal horses were infected, a significantly high proportion of horses with clinical signs of respiratory disease were shedding EHV5. No such association was detected in horses infected with EHV1, -4 and -2.

The increased proportion of horses shedding EHV5 among diseased horses in this study may reflect the contribution of EHV5 to respiratory disease. Alternatively, this shedding may have been reactivated as a consequence of a respiratory disease-associated inflammatory response. The spectrum of clinical disease (or lack of) following gammaherpesvirus infections in horses may be due to a range of factors including virus strain and load, host factors such as age [11,53], and immune responses [54]. Each of these complex factors has been explored in several studies and may help to explain the lack of disease seen in many infected horses. EHV5 is persistently associated with EMPF while it is also regularly detected in both clinically normal and diseased horses [11,13,14,15,16,25,27,31,32,33]. This study showed a significant difference in the proportion of horses shedding EHV5 in the diseased group, such that the odds of disease signs in EHV5-positive horses were twice that of EHV5-negative horses. This difference may be the result of lytic EHV5 infection causing the clinical signs, or that EHV5 is reactivated by infection/inflammation by another agent. B-lymphocytes are a latent reservoir for EHV2 and EHV5, and other sites may exist which have not yet been identified [44,55,56,57]. However, simple reactivation of shedding via B-lymphocytes recruited to these sites does not account for the difference in the clinical associations of EHV5 and EHV2 in this study. Other studies have also shown a protective effect of EHV2 against *Rhodococcus equi* infection [58]. Whether EHV2 and EHV5 each occupy distinct niches within the respiratory tract, or whether each recruit different types of inflammatory cells that might be protective or immunopathogenic, remains unknown.

The higher incidence of EHV2 in non-diseased horses in this study is consistent with those of previous studies and continues to confound our understanding of the role of this virus, if any, in equine respiratory tract disease. The prevalence of EHV2 infection in large numbers of clinically normal horses has been widely reported [13,14,15,16,26,27]; however, several studies have identified associations between EHV2 infection and mild respiratory disease, particularly in foals [11,13,14,17,59,60]. 

Quantification of gammaherpesvirus shedding may enable an association to be made between viral load and clinical disease. In humans, an age-range-specific correlation exists between the levels of the gammaherpesvirus Epstein–Barr virus (EBV) in blood, and the presence of clinical disease [61]; however, there is currently little evidence in this or other studies to support an association with acute respiratory disease and gammaherpesvirus load in horses [11,58,62]. Multiple factors are likely to be required for gammaherpesvirus-mediated disease in horses, rather than solely lytic infections. Alternatively, nasal samples may not be the most appropriate samples as predictors for clinical respiratory disease. This is supported by a recent publication linking high viral loads of EHV5 in bronchoalveolar lavage fluid to EMPF [18].

The detection of alphaherpesviruses is reported in a minority of horses within most populations [17,19,20,63,64,65,66]. Five horses without signs of disease were shedding high levels of either alphaherpesvirus EHV1 or -4, consistent with the “cycle of silent herpesvirus shedding” and spread [63,66,67]. The reactivation of latent alphaherpesvirus infection is associated with subclinical viral shedding [20] and can occur following stressful events such as social re-grouping, weaning and long-distance transport [35,65,68,69]. Despite these factors, the levels of detection of equine herpesviruses in this study population were consistent with other studies that have reported ranges of 0–10% for the alphaherpesviruses and 0–100% for the gammaherpesviruses [12,14,16,20,51,63,65,67,70]. The reactivation of latent herpesviruses following a single immunosuppressive event may explain the detection of multiple herpesviruses. This phenomenon has been documented in humans with prolonged sepsis [71]. Shedding of multiple EHVs was detected in 14% (57/407) of horses. Four of the six horses (67%) infected with the alphaherpesviruses EHV1 and -4 were infected by either another alpha- or a gammaherpesvirus(es).

Although Victoria remained free of EIV during Australia’s only recorded EI outbreak, field staff faced logistical challenges and were often time poor. However, samples were successfully collected, and testing was not compromised, ensuring that EIV could be ruled out in all samples analysed. The lack of comprehensive histories and clinical detail for every horse including age, vaccination status and time-course of clinical disease may have limited the analysis of data. The inclusion of the 38 horses of unknown clinical status was made to assist the determination of overall prevalence. A separate analysis of these horses did not show any statistical difference in the proportion of EHV infection in these horses compared with those of known status (diseased and non-diseased).

## 5. Conclusions

The clinical significance of respiratory gammaherpesvirus infections in horses remains to be determined; however, this survey adds to the mounting body of evidence associating EHV5 with equine respiratory disease. The task of identifying a definitive role of the equine gammaherpesviruses as the cause of respiratory disease on a case-by-case basis remains challenging, since the precise role of both EHV2 and -5 and their relation to clinical disease is likely to be complex and remains to be elucidated for these enigmatic viruses.

## Figures and Tables

**Figure 1 animals-11-03418-f001:**
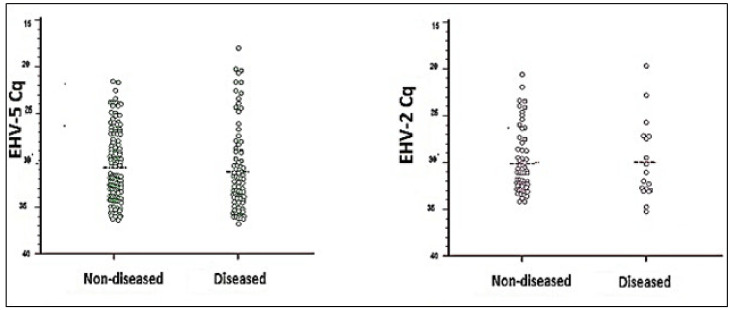
Quantification cycles (Cq) values considered positive for Equine herpesvirus 2 and -5 in nasal swabs of horses with and without clinical signs of disease (diseased and non-diseased). The horizontal line indicates the mean Cq for each group, none of which were statistically different between groups.

**Table 1 animals-11-03418-t001:** Clinical status of horses and detection of. Equine Herpesvirus 1, -4, -2 and -5 from nasal swabs.

Virus Detected	Respiratory Disease Signs			
	Negative	Positive	Not Recorded	Total
EHV-1 only	1	0	0	1
EHV-2 only	29	4	0	33
EHV-5 only	105	70	22	197
EHV-1 and EHV-4	2	0	0	2
EHV-2 and EHV-5	31	14	4	49
EHV-4 and EHV-5	0	1	1	2
EHV-2, EHV-5 and EHV-4	1	0	0	1
No detection	80	31	11	122
Total	249	120	38	407

## Data Availability

Not applicable.

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
