# Peer review of "Association of Equine Herpesvirus 5 with Mild Respiratory Disease in a Survey of EHV1, -2, -4 and -5 in 407 Australian Horses"

_animals, 2021, doi:10.3390/ani11123418_

Round 1
Reviewer 1 Report
This manuscript takes the opportunity to explore the prevalence of EHV-2 and EHV-5 in equine nasal swabs which were collected as part of equine influenza screening in Australia in 2007. Where the data were available, the respiratory disease status was also recorded. Aside from the association between EHV-5 and equine multi-nodular pulmonary fibrosis, whether these gamma herpesviruses are responsible for respiratory disease is an on-going debate, so the generation of more data from field samples is worthwhile and adds knowledge.
The manuscript is neatly presented with clear tables and figures. My main comment relates to Table 1. First, some of the "totals" don't add up and second many of the numbers, particularly about the gamma herpesviruses, bear no obvious relation to the description of results in the text, so checking is needed here please and / or a better explanation of why the numbers are so different. Otherwise, only minor corrections to the text are required (file attached). The conclusions are justified. I enjoyed reading the paper too!
Throughout, please be consistent in the use of either EHV-1 or EHV1 etc eg L37 EHV-5 versus L42 EHV5
L16 valuable information in terms of
L17 were detected positive for the gamma….
L19 in horses;
L22 This study has added tom our
L26 Equine herpesviruses (EHVs)
L30 Nasal swabs were collected from
L40 gamma herpesvirus
L108 replace Dynon et al 2001 with numerical format
L117 Equine herpesvirus 1 can be abbreviated to EHV1
Table 1 – it’s unclear how the total was reached in the first and final rows – the numbers don’t add up. The description in the text also doesn’t align. Please correct these errors.
L160 Section 3.2 – for consistency, this should be Equine alphaherpesvirus infections, EHV1 and -4 because Section 3.3 is titled Equine gammaherpesvirus infections EHV2 and -5
L238 the odds of disease signs in EHV5 positive horses were twice that ….
L249 The higher incidence…. is consistent….. and continues to confound…
L259 with acute respiratory disease and gammaherpesvirus….
L288 known statuses
L292 remains yet to be determined
L296 and remains yet to be elucidated
Author Response
Thank you appreciated your comments and the edits have certainly improved the clarity and consistency of the manuscript -we apologise for many of these oversights
Amended table is in the attached file with responses and below
Throughout, please be consistent in the use of either EHV-1 or EHV1 etc eg L37 EHV-5 versus L42 EHV5
Sure we chose to not hyphenate (eg EHV1) throughout the text however there were some later oversights that said some confusion may lay in the fact that I would hyphenate if after prefixing it Eg EHV1, -4,-2,-5 However the default will be no hyphen for each EHV
L16 valuable information in terms of Amended thanks
L17 were detected positive for the gamma…. Amended thanks
L19 in horses; inserted thanks
L22 This study has added tom our done oops
L26 Equine herpesviruses (EHVs) amended
L30 Nasal swabs were collected from added
L40 gamma herpesvirus amended
L108 replace Dynon et al 2001 with numerical format done
L117 Equine herpesvirus 1 can be abbreviated to EHV1done
Table 1 – it’s unclear how the total was reached in the first and final rows – the numbers don’t add up. The description in the text also doesn’t align. Please correct these errors.-rookie error amended apologies for the error in final table format error corrected in this amended table
Table 1: Clinical status of horses and detection of EHV1, -4, -2 and -5 from nasal swabs
|
Virus detected |
Respiratory disease signs |
|||
|
Positive |
Negative |
Not Recorded |
Total |
|
|
EHV-1 only |
1 |
0 |
0 |
1 |
|
EHV-2 only |
29 |
4 |
0 |
33 |
|
EHV-5 only |
105 |
70 |
22 |
197 |
|
EHV-1 and EHV-4 |
2 |
0 |
0 |
2 |
|
EHV-2 and EHV-5 |
31 |
14 |
4 |
49 |
|
EHV-4 and EHV-5 |
0 |
1 |
1 |
2 |
|
EHV-2, EHV-5 and EHV-4 |
1 |
0 |
0 |
1 |
|
No detection |
80 |
31 |
11 |
122 |
|
Total |
249 |
120 |
38 |
407 |
L160 Section 3.2 – for consistency, this should be Equine alphaherpesvirus infections, EHV1 and -4 because Section 3.3 is titled Equine gammaherpesvirus infections EHV2 and -5-amended thanks
L238 the odds of disease signs in EHV5 positive horses were twice that …. thanks amended poor grammar
L249 The higher incidence…. is consistent….. and continues to confound… thanks amended poor grammar
L259 with acute respiratory disease and gammaherpesvirus….inserted disease
L288 known statuses amended
L292 remains yet to be determined deleted
L296 and remains yet to be elucidated deleted

Reviewer 2 Report
The manuscript describes a study to determine the prevalence of EHV-1,2,4,5 and the possible association of these viruses to clinical infection among horses in Victoria, Australia. The manuscript, as the authors started, adds to our understanding of the prevalence of these important viruses among horse population in general and in Australia in particular. The design of the study is straight forward and only dependent on the use of qPCR in viral DNA detection in nasal swabs. qPCR is the most sensitive assay to determine EHV shedding in horses. The outcomes of the study are consistent with previous reports that described the prevalence of EHVs among horses in different places in the world; indicating that these viruses, more or less, are behaving the same.
the main comment on the current study: why the authors did not try to sequence some of the positive samples in order to compare the detected viruses with other strains? this would be very helpful not only for the current study but also to shed light on the circulating viruses in the area.
Author Response
We totally agree with the suggestion that sequencing may have added to this study and in hindsight would have performed this particularly in light of the vagaries of infection by some herpesviruses despite lack of obvious clinical signs. We noted indeed results of the melt curve analysis for the equine gammaherpesviruses suggested there was some sequence variation as might have been expected.
Unfortunately there were several limiting factors regarding sequencing, swabs were collected by tens of field staff submitting samples for EIV exclusion-samples were submitted and stored in a wide range of conditions that were generally adequate for basic PCR detection however may have been an issue for sequencing. This is particularly relevant as the work was performed several years ago prior to the more reliable and cost effective virus sequencing methods that have since become a practical and readily available tool.